# Pseudo-LiDAR++:
# Accurate Depth for 3D Object Detection in Autonomous Driving

**Yurong You**[*1], **Yan Wang**[*1], **Wei-Lun Chao**[*2], **Divyansh Garg**[1], **Geoff Pleiss**[1],
**Bharath Hariharan**[1], **Mark Campbell**[1], and **Kilian Q. Weinberger**[1]
[1]Cornell University, Ithaca, NY      [2]The Ohio State University, Columbus, OH
`{yy785, yw763, dg595, gp346, bh497, mc288, kqw4}@cornell.edu`
`chao.209@osu.edu`

## Abstract

Detecting objects such as cars and pedestrians in 3D plays an indispensable role in autonomous driving. Existing approaches largely rely on expensive LiDAR sensors for accurate depth information. While recently pseudo-LiDAR has been introduced as a promising alternative, at a much lower cost based solely on stereo images, there is still a notable performance gap. In this paper we provide substantial advances to the pseudo-LiDAR framework through improvements in stereo depth estimation. Concretely, we adapt the stereo network architecture and loss function to be more aligned with accurate depth estimation of faraway objects — currently the primary weakness of pseudo-LiDAR. Further, we explore the idea to leverage cheaper but extremely sparse LiDAR sensors, which alone provide insufficient information for 3D detection, to de-bias our depth estimation. We propose a depth-propagation algorithm, guided by the initial depth estimates, to diffuse these few exact measurements across the entire depth map. We show on the KITTI object detection benchmark that our combined approach yields substantial improvements in depth estimation and stereo-based 3D object detection — outperforming the previous state-of-the-art detection accuracy for faraway objects by $40\%$. Our code is available at `https://github.com/mileyan/Pseudo_Lidar_V2`.

## 1 Introduction

Safe driving in autonomous cars requires accurate 3D detection and localization of cars, pedestrians and other objects. This in turn requires accurate depth information, which can be obtained from LiDAR (Light Detection And Ranging) sensors. Although highly precise and reliable, LiDAR sensors are notoriously expensive: a 64-beam model can cost around $75,000 (USD)[1]. The alternative is to measure depth through inexpensive commodity cameras. However, in spite of recent dramatic progress in stereo-based 3D object detection brought by pseudo-LiDAR (Wang et al., 2019a), a significant performance gap remains especially for faraway objects (which we want to detect early to allow time for reaction). The trade-off between affordability and safety creates an ethical dilemma.

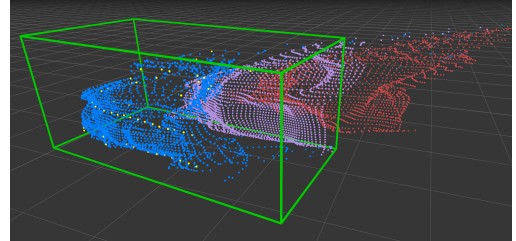

Figure 1: **An illustration of our proposed depth estimation and correction method.** The green box is the ground truth location of the car in the KITTI dataset. The red points are obtained with a stereo disparity network. Purple points, obtained with our stereo depth network (SDN), are much closer to the truth. After depth propagation (blue points) with a few (yellow) LiDAR measurements the car is squarely inside the green box. (One floor square is 1m×1m.)

---

[*]Equal contributions

[1]The information is obtained from the automotive LiDAR market report: `http://www.woodsidecap.com/wp-content/uploads/2018/04/Yole_WCP-LiDAR-Report_April-2018-FINAL.pdf`

In this paper we propose a possible solution to this remaining challenge that combines insights from both perspectives. We observe that the higher 3D object localization error of stereo-based systems, compared to LiDAR-based ones, stems entirely from the higher error in depth estimation (after the 3D point cloud is obtained the two approaches are identical (Wang et al., 2019a)). Importantly, this error is not random but *systematic*: we observe that stereo methods do indeed *detect* objects with high reliability, yet they estimate the depth of the *entire* object as either too far or too close. See Figure 1 for an illustration: the red stereo points capture the car but are shifted by about 2m completely outside the ground-truth location (green box). If we can *de-bias* these depth estimates it should be possible to obtain accurate 3D localization even for distant objects without exorbitant costs.

We start by revisiting the depth estimation routine embedded at the heart of state-of-the-art stereo-based 3D detection approach (Wang et al., 2019a). A major contributor to the systematic depth bias comes from the fact that depth is typically not computed directly. Instead, one first estimates the *disparity* — the horizontal shift of a pixel between the left and right images — and then *inverts* it to obtain pixel-wise depth. While the use of deep neural networks has largely improved disparity estimation (Chang & Chen, 2018; Cheng et al., 2018; Mayer et al., 2016; Wang et al., 2019b), designing and learning the networks to optimize the accuracy of *disparity estimation* simply over-emphasizes nearby objects due to the reciprocal transformation. For instance, a unit disparity error (in pixels) for a 5-meter-away object means a 10cm error in depth: the length of a side mirror. The same disparity error for a 50-meter-away object, however, becomes a 5.8m error in depth: the length of an entire car. Penalizing both errors equally means that the network spends more time correcting subtle errors on nearby objects than gross errors on faraway objects, resulting in degraded depth estimates and ultimately poor detection and localization for faraway objects. We thus propose to adapt the stereo network architecture and loss function for direct depth estimation. Concretely, the cost volume that fuses the left-right images and the subsequent 3D convolutions are the key components in stereo networks. Taking the central assumption of convolutions — all neighborhoods can be operated in an identical manner — we propose to construct the cost volume on the grid of depth rather than disparity, enabling 3D convolutions and the loss function to perform exactly on the right scale for depth estimation. We refer to our network as stereo depth network (SDN). See Figure 1 for a comparison of 3D points obtained with SDN (purple) and disparity estimation (red).

Although our SDN improves the depth estimates significantly, stereo images are still inherently 2D and it is unclear if they can ever match the accuracy and reliability of a true 3D LiDAR sensor. Although LiDAR sensors with 32 or 64 beams are expensive, LiDAR sensors with only 4 beams are two orders of magnitude cheaper[2] and thus easily affordable. The 4 laser beams are very sparse and ill-suited to capture 3D object shapes by themselves, but if paired with stereo images they become the ideal tool to de-bias our dense stereo depth estimates: a single high-precision laser beam may inform us how to correct the depth of an entire car or pedestrian in its path. To this end, we present a novel depth-propagation algorithm, inspired by graph-based manifold learning (Weinberger et al., 2005; Roweis & Saul, 2000; Xiaojin & Zoubin, 2002). In a nutshell, we connect our estimated 3D stereo point cloud locally by a nearest neighbor graph, such that points corresponding to the same object will share many local paths with each other. We match the few but exact LiDAR measurements first with pixels (irrespective of depth) and then with their corresponding 3D points to obtain accurate depth estimates for several nodes in the graph. Finally, we propagate this exact depth information along the graph using a label diffusion mechanism — resulting in a *dense and accurate depth map* at *negligible cost*. In Figure 1 we see that the few (yellow) LiDAR measurements are sufficient to position almost all final (blue) points of the entire car within the green ground truth box.

We conduct extensive empirical studies of our approaches on the KITTI object detection benchmark (Geiger et al., 2012; 2013) and achieve remarkable results. With solely stereo images, we outperform the previous state of the art (Wang et al., 2019a) by $10\%$. Further adding a cheap 4-beam LiDAR brings another $27\%$ relative improvement — on some metrics, our approach is nearly on par with those based on a 64-beam LiDAR but can potentially save $95\%$ in cost.

---

[2]The Ibeo Wide Angle Scanning (ScaLa) sensor with 4 beams costs $600 (USD). In this paper we simulate the 4-beam LiDAR signal on KITTI benchmark (Geiger et al., 2012) by sparsifying the original 64-beam signal.

## 2 BACKGROUND

**3D object detection.** Most work on 3D object detection operates on 3D point clouds from LiDAR as input (Li, 2017; Li et al., 2016; Meyer et al., 2019b; Yang et al., 2018a; Du et al., 2018; Shi et al., 2019; Engelcke et al., 2017; Yan et al., 2018; Lang et al., 2019). Frustum PointNet (Qi et al., 2018) applies PointNet (Qi et al., 2017a;b) to the points directly, while Voxelnet (Zhou & Tuzel, 2018) quantizes them into 3D grids. For street scenes, several work finds that processing points from the bird's-eye view can already capture object contours and locations (Chen et al., 2017; Yang et al., 2018b; Ku et al., 2018). Images have also been used, but mainly to supplement LiDAR (Meyer et al., 2019a; Xu et al., 2018; Liang et al., 2018; Chen et al., 2017; Ku et al., 2018). Early work based solely on images — mostly built on the 2D frontal-view detection pipeline (Ren et al., 2015; He et al., 2017; Lin et al., 2017) — fell far behind in localizing objects in 3D (Li et al., 2019a; Xiang et al., 2015; 2017; Chabot et al., 2017; Mousavian et al., 2017; Chen et al., 2015; Xu & Chen, 2018; Chen et al., 2016; Pham & Jeon, 2017; Chen et al., 2018)[3].

**Pseudo-LiDAR.** This gap has been reduced significantly recently with the introduction of the pseudo-LiDAR framework proposed in (Wang et al., 2019a). This framework applies a drastically different approach from previous image-based 3D object detectors. Instead of directly detecting the 3D bounding boxes from the frontal view of a scene, pseudo-LiDAR begins with image-based depth estimation, predicting the depth $Z(u, v)$ of each image pixel $(u, v)$. The resulting depth map $Z$ is then back-projected into a 3D point cloud: a pixel $(u, v)$ will be transformed to $(x, y, z)$ in 3D by

$$z = Z(u, v), \qquad x = \frac{(u - c_U) \times z}{f_U}, \qquad y = \frac{(v - c_V) \times z}{f_V}, \tag{1}$$

where $(c_U, c_V)$ is the camera center and $f_U$ and $f_V$ are the horizontal and vertical focal length. The 3D point cloud is then treated exactly as LiDAR signal — any LiDAR-based 3D detector can be applied seamlessly. By taking the state-of-the-art algorithms from both ends (Chang & Chen, 2018; Ku et al., 2018; Qi et al., 2018), pseudo-LiDAR obtains the highest image-based performance on the KITTI object detection benchmark (Geiger et al., 2012; 2013). Our work builds upon this framework.

**Stereo disparity estimation.** Pseudo-LiDAR relies heavily on the quality of depth estimation. Essentially, if the estimated pixel depths match those provided by LiDAR, pseudo-LiDAR with any LiDAR-based detector should be able to achieve the same performance as that obtained by applying the same detector to the LiDAR signal. According to (Wang et al., 2019a), depth estimation from stereo pairs of images (Mayer et al., 2016; Yamaguchi et al., 2014; Chang & Chen, 2018) are more accurate than that from monocular (i.e., single) images (Fu et al., 2018; Godard et al., 2017) for 3D object detection. We therefore focus on stereo depth estimation, which is routinely obtained from estimating disparity between images.

A disparity estimation algorithm takes a pair of left-right images $I_l$ and $I_r$ as input, captured from a pair of cameras with a horizontal offset (i.e., baseline) $b$. Without loss of generality, we assume that the algorithm treats the left image, $I_l$, as reference and outputs a disparity map $D$ recording the horizontal disparity to $I_r$ for each pixel $(u, v)$. Ideally, $I_l(u, v)$ and $I_r(u, v + D(u, v))$ will picture the same 3D location. We can therefore derive the depth map $Z$ via the following transform,

$$Z(u, v) = \frac{f_U \times b}{D(u, v)} \quad (f_U: \text{horizontal focal length}). \tag{2}$$

A common pipeline of disparity estimation is to first construct a 4D disparity cost volume $C_{\text{disp}}$, in which $C_{\text{disp}}(u, v, d, :)$ is a feature vector that captures the pixel difference between $I_l(u, v)$ and $I_r(u, v + d)$. It then estimates the disparity $D(u, v)$ for each pixel $(u, v)$ according to the cost volume $C_{\text{disp}}$. One basic algorithm is to build a 3D cost volume with $C_{\text{disp}}(u, v, d) = \|I_l(u, v) - I_r(u, v + d)\|_2$ and determine $D(u, v)$ as $\arg \min_d C_{\text{disp}}(u, v, d)$. Advanced algorithms exploit more robust features in constructing $C_{\text{disp}}$ and perform structured prediction for $D$. In what follows, we give an introduction of PSMNet (Chang & Chen, 2018), a state-of-the-art algorithm used in (Wang et al., 2019a).

PSMNet begins with extracting deep feature maps $h_l$ and $h_r$ from $I_l$ and $I_r$, respectively. It then constructs $C_{\text{disp}}(u, v, d, :)$ by concatenating features of $h_l(u, v)$ and $h_r(u, v + d)$, followed by layers

---

[3]Recently, Srivastava et al. (2019) proposed to lift 2D monocular images to 3D representations (e.g., bird's-eye view (BEV) images) and achieved promising monocular-based 3D object detection results.

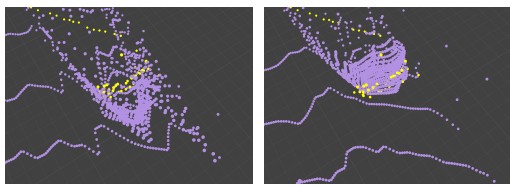
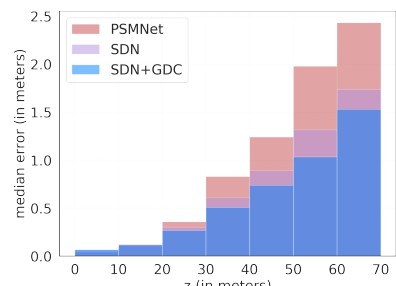

Figure 3: **Disparity cost volume (left) vs. depth cost volume (right).** The figure shows the 3D points obtained from LiDAR (yellow) and stereo (purple) corresponding to a car in KITTI, seen from the bird's-eye view (BEV). Points from the disparity cost volume are stretched out and noisy; while points from the depth cost volume capture the car contour faithfully.

Figure 4: **Depth estimation errors.** We compare depth estimation error on 3,769 KITTI validation images, taking 64-beam LiDAR depths as ground truths. We separate pixels according to their true depths (z). See the text and appendix for details.

of 3D convolutions. The resulting 3D tensor $S_{\text{disp}}$, with the feature channel size ending up being one, is then used to derive the pixel disparity via the following weighted combination,

$$D(u, v) = \sum_d \text{softmax}(-S_{\text{disp}}(u, v, d)) \times d, \qquad (3)$$

where $\text{softmax}$ is performed along the 3$^{\text{rd}}$ dimension of $S_{\text{disp}}$. PSMNet can be learned end-to-end, including the image feature extractor and 3D convolution kernels, to minimize the disparity error

$$\sum_{(u,v) \in \mathcal{A}} \ell(D(u, v) - D^\star(u, v)), \qquad (4)$$

where $\ell$ is the smooth L1 loss, $D^\star$ is the ground truth map, and $\mathcal{A}$ contains pixels with ground truths.

## 3 STEREO DEPTH NETWORK (SDN)

A stereo network designed and learned to minimize the disparity error (cf. Equation 4) may over-emphasize nearby objects with smaller depths and therefore perform poorly in estimating depths for faraway objects. To see this, note that Equation 2 implies that for a given error in disparity $\delta D$, the error in depth $\delta Z$ increases *quadratically* with depth:

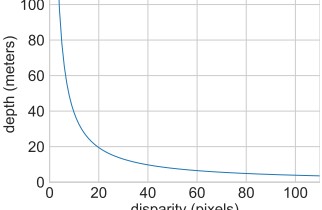

$$Z \propto \frac{1}{D} \Rightarrow \delta Z \propto \frac{1}{D^2} \delta D \Rightarrow \delta Z \propto Z^2 \delta D. \qquad (5)$$

The middle term is obtained by differentiating $Z(D)$ w.r.t. $D$. In particular, using the settings on the KITTI dataset (Geiger et al., 2012; 2013), a single pixel error in disparity implies only a 0.1m error in depth at a depth of 5 meters, but a 5.8m error at a depth of 50 meters. See Figure 2 for a mapping from disparity to depth.

Figure 2: **The disparity-to-depth transform.** We set $f_U = 721$ (in pixels) and $b = 0.54$ (in meters) in Equation 2, which are the typical values used in the KITTI dataset.

**Depth Loss.** We propose two changes to adapt stereo networks for direct depth estimation. First, we learn stereo networks to directly optimize the depth loss

$$\sum_{(u,v) \in \mathcal{A}} \ell(Z(u, v) - Z^\star(u, v)). \qquad (6)$$

$Z$ and $Z^\star$ can be obtained from $D$ and $D^\star$ using Equation 2. The change from the disparity loss to the depth loss corrects the disproportionally strong emphasis on tiny depth errors of nearby objects — a necessary but still insufficient change to overcome the problems of disparity estimation.

**Depth Cost Volume.** To facilitate accurate depth learning (rather than disparity) we need to further address the internals of the depth estimation pipeline. A crucial source of error is the 3D convolutions within the 4D disparity cost volume, where the same kernels are applied for the entire cost volume. This is highly problematic as it implicitly assumes that the effect of a convolution is homogeneous

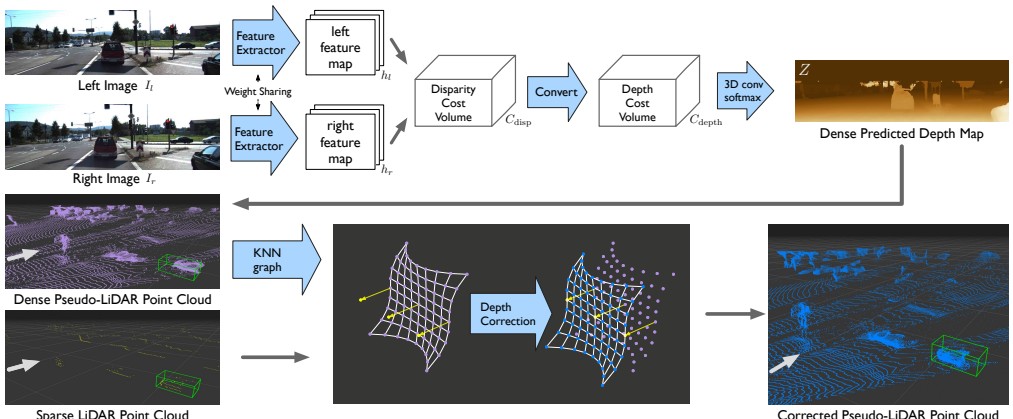

Figure 5: **The whole pipeline of improved stereo depth estimation:** (top) the stereo depth network (SDN) constructs a depth cost volume from left-right images and is optimized for direct depth estimation; (bottom) the graph-based depth correction algorithm (GDC) refines the depth map by leveraging sparser LiDAR signal. The gray arrows indicates the observer's view point. We superimpose the (green) ground-truth 3D box of a car, the same one in Figure 1. The corrected points (blue; bottom right) are perfectly located inside the ground truth box.

throughout — which is clearly violated by the reciprocal depth to disparity relation (Figure 2). For example, it may be completely appropriate to locally smooth two neighboring pixels with disparity 85 and 86 (changing the depth by a few cm to smooth out a surface), whereas applying the same kernel for two pixels with disparity 5 and 6 could easily move the 3D points by 10m or more.

Taking this insight and the central assumption of convolutions — all neighborhoods can be operated upon in an identical manner — into account, we propose to instead construct the depth cost volume $C_{\text{depth}}$, in which $C_{\text{depth}}(u, v, z, :)$ will encode features describing how likely the depth $Z(u, v)$ of pixel $(u, v)$ is $z$. The subsequent 3D convolutions will then operate on the grid of depth, rather than disparity, affecting neighboring depths identically, independent of their location. The resulting 3D tensor $S_{\text{depth}}$ is then used to predict the pixel depth similar to Equation 3

$$Z(u, v) = \sum_z \text{softmax}(-S_{\text{depth}}(u, v, z)) \times z.$$

We construct the new depth volume, $C_{\text{depth}}$, based on the intuition that $C_{\text{depth}}(u, v, z, :)$ and $C_{\text{disp}}\left(u, v, \frac{f_U \times b}{z}, :\right)$ should lead to equivalent "cost". To this end, we apply a bilinear interpolation to construct $C_{\text{depth}}$ from $C_{\text{disp}}$ using the depth-to-disparity transform in Equation 2. Specifically, we consider disparity in the range of $[0, 191]$ following PSMNet (Chang & Chen, 2018), and consider depth in the range of $[1\text{m}, 80\text{m}]$ and set the grid of depth in $C_{\text{depth}}$ to be 1m. Figure 5 (top) depicts our stereo depth network (SDN) pipeline. Crucially, all convolution operations are operated on $C_{\text{depth}}$ exclusively. Figure 4 compares the median values of absolute depth estimation errors using the disparity cost volume (i.e., PSMNet) and the depth cost volume (SDN) (see subsection D.5 for detailed numbers). As expected, for faraway depth, SDN leads to drastically smaller errors with only marginal increases in the very near range (which disparity based methods over-optimize). See the appendix for the detailed setup and more discussions.

## 4 DEPTH CORRECTION

Our SDN significantly improves depth estimation and more precisely renders the object contours (see Figure 3). However, there is a fundamental limitation in stereo because of the discrete nature of pixels: the disparity, being the difference in the horizontal coordinate between corresponding pixels, has to be *quantized* at the level of individual pixels while the depth is *continuous*. Although the quantization error can be alleviated with higher resolution images, the computational depth prediction cost scales *cubically* with resolution— pushing the limits of GPUs in autonomous vehicles.

We therefore explore a hybrid approach by leveraging a cheap LiDAR with extremely sparse (e.g., 4 beams) but accurate depth measurements to *correct* this bias. We note that such sensors are too

*sparse* to capture object shapes and cannot be used alone for detection. However, by projecting the LiDAR points into the image plane we obtain exact depths on a small portion of "landmark" pixels.

We present a graph-based depth correction (GDC) algorithm that effectively combines the *dense* stereo depth that has rendered object shapes and the *sparse* accurate LiDAR measurements. Conceptually, we expect the corrected depth map to have the following properties: globally, landmark pixels associated with LiDAR points should possess the exact depths; locally, object shapes captured by neighboring 3D points, back-projected from the input depth map (cf. Equation 1), should be preserved. Figure 5 (bottom) illustrates the algorithm.

**Input Matching.** We take as input the two point clouds from LiDAR (L) and Pseudo-LiDAR (PL) by stereo depth estimation. The latter is obtained by converting pixels $(u, v)$ with depth $z$ to 3D points $(x_u, y_v, z)$. First, we characterize the local shapes by the directed K-nearest-neighbor (KNN) graph in the PL point cloud (using accelerated KD-Trees (Shevtsov et al., 2007)) that connects each 3D point to its KNNs with appropriate weights. Similarly, we can project the 3D LiDAR points onto pixel locations $(u, v)$ and match them to corresponding 3D stereo points. Without loss of generality, we assume that we are given "ground truth" LiDAR depth for the first $n$ points and no ground truth for the remaining $m$ points. We refer to the 3D stereo depth estimates as $Z \in \mathbb{R}^{n+m}$ and the LiDAR depth ground-truth as $G \in \mathbb{R}^n$.

**Edge weights.** To construct the KNN graph in 3D we ignore the LiDAR information on the first $n$ points and only use their predicted stereo depth in $Z$. Let $\mathcal{N}_i$ denote the set of $k$ neighbors of the $i^{th}$ point. Further, let $W \in \mathbb{R}^{(n+m) \times (n+m)}$ denote the weight matrix, where $W_{ij}$ denotes the edge-weight between points $i$ and $j$. Inspired by prior work in manifold learning (Roweis & Saul, 2000; Weinberger et al., 2005) we choose the weights to be the coefficients that reconstruct the depth of any point from the depths of its neighbors in $\mathcal{N}_i$. We can solve for these weights with the following constrained quadratic optimization problem:

$$W = \arg\min_W \|Z - WZ\|_2^2, \qquad \text{s.t. } W\mathbf{1} = \mathbf{1} \text{ and } W_{ij} = 0 \text{ if } j \notin \mathcal{N}_i. \qquad (7)$$

Here $\mathbf{1} \in \mathbb{R}^{n+m}$ denotes the all-ones vector. As long as we pick $k > 3$ and the points are in general position there are infinitely many solutions that satisfy $Z = WZ$, and we pick the solution with the minimum $L_2$ norm (obtained with slight $L_2$ regularization).

**Depth Correction.** Let us denote the corrected depth values as $Z' \in \mathbb{R}^{n+m}$, with $Z' = [Z'_L; Z'_{PL}]$ and $Z'_L \in \mathbb{R}^n$ and $Z'_{PL} \in \mathbb{R}^m$, where $Z'_L$ are the depth values of points with LiDAR ground-truth and $Z'_{PL}$ otherwise. For the $n$ points with LiDAR measurements we update the depth to the (ground truth) values $Z'_L = G$. We then solve for $Z'_{PL}$ given $G$ and the weighted KNN graph encoded in $W$. Concretely, we update the remaining depths $Z'_{PL}$ such that the depth of any point $i$ can still be be reconstructed with high fidelity as a weighted sum of its KNNs' depths using the learned weights $W$; i.e. if point $i : 1 \le i \le n$ is moved to its new depth $G_i$, then its neighbors in $\mathcal{N}_i$ must also be corrected such that $G_i \approx \sum_{j \in \mathcal{N}_i} W_{ij} Z'_j$. Further, the neighbors' neighbors must be corrected and the depth of the few $n$ points propagates across the entire graph. We can solve for the final $Z'$ directly with another quadratic optimization:

$$Z' = \arg\min_{Z'} \|Z' - WZ'\|^2, \qquad \text{s.t. } Z'_{1:n} = G. \qquad (8)$$

To illustrate the correction process, imagine the simplest case where the depth of only a single point ($n = 1$) is updated to $G_1 = Z_1 + \delta$. A new optimal depth for Equation 8 is to move all the remaining points similarly, i.e. $Z' = Z + \mathbf{1}\delta$: as $Z = WZ$ and $W\mathbf{1} = \mathbf{1}$ we must have $W(Z + \mathbf{1}\delta) = Z + \mathbf{1}\delta$. In the setting with $n > 1$, the least-squares loss ensures a soft diffusion between the different LiDAR depth estimates. Both optimization problems in Equation 7 and Equation 8 can be solved exactly and efficiently with sparse matrix solvers. We summarize the procedure as an algorithm in the appendix.

From the view of graph-based manifold learning, our GDC algorithm is reminiscent of locally linear embeddings (Roweis & Saul, 2000) with landmarks to guide the final solution (Weinberger et al., 2005). Figure 1 illustrates vividly how the initial 3D point cloud from SDN (purple) of a car in the KITTI dataset is corrected with a few sparse LiDAR measurements (yellow). The resulting points (blue) are right inside the ground-truth box and clearly show the contour of the car. Figure 4 shows the additional improvement from the GDC (blue) over the pure SDN depth estimates (see subsection D.5 for detailed numbers). The error (calculated only on non-landmark pixels) is corrected over the entire image where many regions have no LiDAR measurements. This is because that the pseudo-LiDAR point cloud is sufficiently dense and we choose $k$ to be large enough (in practice, we use $k = 10$)

Table 1: **3D object detection results on KITTI validation.** We report AP$_{BEV}$ / AP$_{3D}$ (in %) of the **car** category, corresponding to average precision of the bird's-eye view and 3D object detection. We arrange methods according to the input signals: M for monocular images, S for stereo images, L for 64-beam LiDAR, and L# for *sparse 4-beam* LiDAR. PL stands for PSEUDO-LiDAR. *Our* PSEUDO-LiDAR ++ *(PL++) with enhanced depth estimation* — SDN *and* GDC— *are in blue.* Methods with 64-beam LiDAR are in gray. Best viewed in color.

| Detection algo | Input | IoU = 0.5 | | | IoU = 0.7 | | |
|---|---|---|---|---|---|---|---|
| | | Easy | Moderate | Hard | Easy | Moderate | Hard |
| 3DOP | S | 55.0 / 46.0 | 41.3 / 34.6 | 34.6 / 30.1 | 12.6 / 6.6 | 9.5 / 5.1 | 7.6 / 4.1 |
| MLF-STEREO | S | - | 53.7 / 47.4 | - | - | 19.5 / 9.8 | - |
| S-RCNN | S | 87.1 / 85.8 | 74.1 / 66.3 | 58.9 / 57.2 | 68.5 / 54.1 | 48.3 / 36.7 | 41.5 / 31.1 |
| PL: AVOD | S | 89.0 / 88.5 | 77.5 / 76.4 | 68.7 / 61.2 | 74.9 / 61.9 | 56.8 / 45.3 | 49.0 / 39.0 |
| PL: PIXOR$^\star$ | S | 89.0 / - | 75.2 / - | 67.3 / - | 73.9 / - | 54.0 / - | 46.9 / - |
| PL: P-RCNN | S | 88.4 / 88.0 | 76.6 / 73.7 | 69.0 / 67.8 | 73.4 / 62.3 | 56.0 / 44.9 | 52.7 / 41.6 |
| PL++: AVOD | S | 89.4 / 89.0 | 79.0 / 77.8 | 70.1 / 69.1 | 77.0 / 63.2 | 63.7 / 46.8 | 56.0 / 39.8 |
| PL++: PIXOR$^\star$ | S | **89.9** / - | 78.4 / - | 74.7 / - | 79.7 / - | 61.1 / - | 54.5 / - |
| PL++: P-RCNN | S | 89.8 / **89.7** | **83.8 / 78.6** | **77.5 / 75.1** | **82.0 / 67.9** | **64.0 / 50.1** | **57.3 / 45.3** |
| PL++: AVOD | L# + S | 90.2 / 90.1 | **87.7 / 86.9** | 79.8 / 79.2 | 86.8 / 70.7 | 76.6 / 56.2 | 68.7 / 53.4 |
| PL++: PIXOR$^\star$ | L# + S | **95.1** / - | 85.1 / - | 78.3 / - | 84.0 / - | 71.0 / - | 65.2 / - |
| PL++: P-RCNN | L# + S | 90.3 / 90.3 | **87.7 / 86.9** | **84.6 / 84.2** | **88.2 / 75.1** | **76.9 / 63.8** | **73.4 / 57.4** |
| AVOD | L + M | 90.5 / 90.5 | 89.4 / 89.2 | 88.5 / 88.2 | 89.4 / 82.8 | 86.5 / 73.5 | 79.3 / 67.1 |
| PIXOR$^\star$ | L + M | 94.2 / - | 86.7 / - | 86.1 / - | 85.2 / - | 81.2 / - | 76.1 / - |
| P-RCNN | L | 97.3 / 97.3 | 89.9 / 89.8 | 89.4 / 89.3 | 90.2 / 89.2 | 87.9 / 78.9 | 85.5 / 77.9 |

such that the KNN graph is typically connected (or consists of few large connected components). See subsection D.6 for more analysis. For objects such as cars the improvements through GDC are far more pronounced, as these typically are touched by the four LiDAR beams and can be corrected effectively.

## 5 EXPERIMENTS

### 5.1 SETUP

We refer to our combined method (SDN and GDC) for 3D object detection as PSEUDO-LiDAR++ (PL++ in short). To analyze the contribution of each component, we evaluate SDN and GDC independently and jointly across several settings. For GDC we set $k = 10$ and consider adding signal from a (simulated) 4-beam LiDAR, unless stated otherwise.

**Dataset, Metrics, and Baselines.** We evaluate on the KITTI dataset (Geiger et al., 2013; 2012), which contains 7,481 and 7,518 images for training and testing. We follow (Chen et al., 2015) to separate the 7,481 images into 3,712 for training and 3,769 validation. For each (left) image, KITTI provides the corresponding right image, the 64-beam Velodyne LiDAR point cloud, the camera calibration matrices, and the bounding boxes. We focus on 3D object detection and bird's-eye-view (BEV) localization and report results on the *validation set*. Specifically, we focus on the "car" category, following Chen et al. (2017) and Xu et al. (2018). We report average precision (AP) with IoU (Intersection over Union) thresholds at 0.5 and 0.7. We denote AP for the 3D and BEV tasks by AP$_{3D}$ and AP$_{BEV}$. KITTI defines the easy, moderate, and hard settings, in which objects with 2D box heights smaller than or occlusion/truncation levels larger than certain thresholds are disregarded. We compare to four stereo-based detectors: PSEUDO-LiDAR (PL in short) (Wang et al., 2019a), 3DOP (Chen et al., 2015), S-RCNN (Li et al., 2019b), and MLF-STEREO (Xu & Chen, 2018).

**Stereo depth network (SDN).** We use PSMNET (Chang & Chen, 2018) as the backbone for our stereo depth estimation network (SDN). We follow Wang et al. (2019a) to pre-train SDN on the synthetic Scene Flow dataset (Mayer et al., 2016) and fine-tune it on the 3,712 training images of KITTI. We obtain the depth ground truth by projecting the corresponding LiDAR points onto images. We also train a PSMNET in the same way for comparison, which minimizes disparity error.

**3D object detection.** We apply three algorithms: AVOD (Ku et al., 2018), PIXOR (Yang et al., 2018b), and P-RCNN (Shi et al., 2019). All utilize information from LiDAR and/or monocular images. We use the released implementations of AVOD (specifically, AVOD-FPN) and P-RCNN. We implement PIXOR ourselves with a slight modification to include visual information (denoted

Table 2: **Results on the car category on the *test* set.** We compare PL++ (blue) and 64-beam LiDAR (gray), using P-RCNN, and report AP_BEV / AP_3D at IoU=0.7.

| Input signal | Easy | Moderate | Hard |
|---|---|---|---|
| PL++ (SDN) | 75.5 / 60.4 | 57.2 / 44.6 | 53.4 / 38.5 |
| PL++ (SDN + GDC) | 83.8 / 68.5 | 73.5 / 54.7 | 66.5 / 51.2 |
| LiDAR | 89.5 / 85.9 | 85.7 / 75.8 | 79.1 / 68.3 |

Table 3: **Ablation study on depth estimation.** We report AP_BEV / AP_3D (in %) of the **car** category at IoU= 0.7 on KITTI validation. DL: depth loss.

| Stereo depth | Easy | Moderate | Hard |
|---|---|---|---|
| PSMNET | 73.4 / 62.3 | 56.0 / 44.9 | 52.7 / 41.6 |
| PSMNET + DL | 80.1 / 65.5 | 61.9 / 46.8 | 56.0 / 43.0 |
| SDN | **82.0 / 67.9** | **64.0 / 50.1** | **57.3 / 45.3** |

Table 4: **Ablation study on leveraging sparse LiDAR.** We report AP_BEV / AP_3D (in %) of the **car** category at IoU= 0.7 on KITTI validation. L#: 4-beam LiDAR signal alone. SDN + L#: pseudo-LiDAR with depths of landmark pixels replaced by 4-beam LiDAR. The best result of each column is in bold font.

| Stereo depth | Easy | Moderate | Hard |
|---|---|---|---|
| SDN | 82.0 / 67.9 | 64.0 / 50.1 | 57.3 / 45.3 |
| L# | 73.2 / 56.1 | 71.3 / 53.1 | 70.5 / 51.5 |
| SDN + L# | 86.3 / 72.0 | 73.0 / 56.1 | 67.4 / 54.1 |
| SDN + GDC | **88.2 / 75.1** | **76.9 / 63.8** | **73.4 / 57.4** |

Table 5: **Results of pedestrians (top) and cyclists (bottom) on KITTI validation.** We apply F-POINTNET Qi et al. (2018) and report AP_BEV / AP_3D (in %) at IoU= 0.5, following Wang et al. (2019a).

| Stereo depth | Easy | Moderate | Hard |
|---|---|---|---|
| PSMNET | 41.3 / 33.8 | 34.9 / 27.4 | 30.1 / 24.0 |
| SDN | 48.7 / 40.9 | 40.4 / 32.9 | 34.9 / 28.8 |
| SDN + GDC | **63.7 / 53.6** | **53.8 / 44.4** | **46.8 / 38.1** |
| PSMNET | 47.6 / 41.3 | 29.9 / 25.2 | 27.0 / 24.9 |
| SDN | 49.3 / 44.6 | 30.4 / 28.7 | 28.6 / 26.4 |
| SDN + GDC | **65.7 / 60.8** | **45.8 / 40.8** | **42.8 / 38.0** |

as PIXOR⋆). We train all models on the 3,712 training data from scratch by replacing the LiDAR points with pseudo-LiDAR data generated from stereo depth estimation. See the appendix for details.

**Sparser LiDAR.** We simulate sparser LiDAR signal with fewer beams by first projecting the 64-beam LiDAR points onto a 2D plane of horizontal and vertical angles. We quantize the vertical angles into 64 levels with an interval of $0.4°$, which is close to the SPEC of the 64-beam LiDAR. We keep points fallen into a subset of beams to mimic the sparser signal. See the appendix for details.

## 5.2 EXPERIMENTAL RESULTS

**Results on the KITTI val set.** We summarize the main results on KITTI object detection in Table 1. Several important trends can be observed: **1)** Our PL++ with enhanced depth estimations by SDN and GDC yields consistent improvement over PL across all settings; **2)** PL++ with GDC refinement of 4-beam LiDAR (Input: L# + S) performs significantly better than PL++ with only stereo inputs (Input: S); **3)** PL experiences a substantial drop in accuracy from IoU at 0.5 to 0.7 for the *hard* setting. This suggests that while PL detects faraway objects, it mislocalizes them, likely placing them at the wrong depth. This causes the object to be considered a missed detection at higher overlap thresholds. Interestingly, here is where we experience the largest gain — from PL: P-RCNN (AP_BEV = 52.7) to PL++: P-RCNN (AP_BEV = 73.4) with input as L# + S. Note that the majority of the gain comes from GDC, as PL++ with the stereo-only version only improving the score to $57.3$ AP_BEV. **4)** The gap between PL++ and LiDAR is at most $13\%$ AP_BEV, even at the hard setting under IoU at 0.7. **5)** For IoU at 0.5, with the aid of only 4 LiDAR beams, PL++ (SDN + GDC) achieves results comparable to models with 64-beam LiDAR signals.

**Results on the KITTI test set.** Table 2 summarizes results on the car category on the KITTI test set. We see a similar gap between our methods and LiDAR as on the validation set, suggesting that our improvement is not particular to the validation data. Our approach without LiDAR refinement (pure SDN) is placed at the top position among all the image-based algorithms on the KITTI leaderboard.

In the following, we conduct a series of experiments to analyze the performance gain by our approaches and discuss several key observations. *We mainly experiment with* P-RCNN*: we find that the results with* AVOD *and* PIXOR⋆ *follow similar trends and thus include them in the appendix.*

**Depth loss and depth cost volume.** To turn a disparity network (e.g., PSMNET) into SDN, there are two changes: **1)** change the disparity loss into the depth loss; **2)** change the disparity cost volume into the depth cost volume. In Table 3, we uncover the effect of these two changes separately. On the AP_BEV/AP_3D (moderate) metric, the depth loss gives us a $6\%/2\%$ improvement and the depth cost volume brings another $2 \sim 3\%$ gain[4].

---

[4]We note that, the degree of improvement brought by the depth loss and depth cost volume depends on the 3D detector in use. Table 3 suggests that the depth loss provides more gain than the depth cost volume (for P-RCNN). In Table 6, we, however, see that the depth cost volume provides comparable or even bigger gain

**Impact of sparse LiDAR beams.** We leverage 4-beam LiDAR to correct stereo depth using GDC. However, it is possible that gains in 3D object detection come entirely from the new LiDAR sensor and that the stereo estimates are immaterial. In Table 4, we study this question by comparing the detection results against those of models using **1)** sole 4-beam LiDAR point clouds and **2)** pseudo-LiDAR point clouds with depths of landmark pixels replaced by 4-beam LiDAR: i.e., in depth correction, we only correct depths of the landmark pixels without propagation. It can be seen that 4-beam LiDAR itself performs fairly well on locating faraway objects but cannot capture nearby objects precisely, while simply replacing pseudo-LiDAR with LiDAR at the landmark pixels prevents the model from detecting faraway object accurately. In contrast, our proposed GDC method effectively combines the merits of the two signals, achieving superior performance than using them alone.

**Pedestrian and cyclist detection.** For a fair comparison to (Wang et al., 2019a), we apply F-POINTNET (Qi et al., 2018) for detecting pedestrians and cyclists. Table 5 shows the results: our methods significantly boosts the performance.

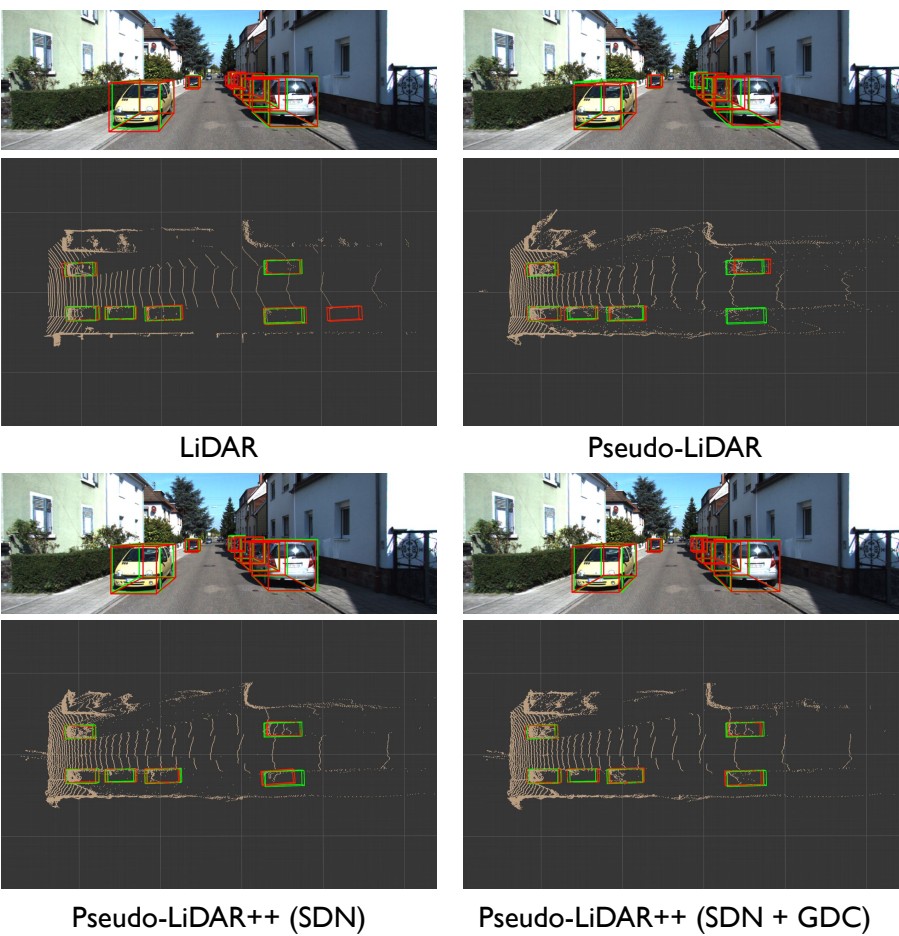

LiDAR       Pseudo-LiDAR

Pseudo-LiDAR++ (SDN)       Pseudo-LiDAR++ (SDN + GDC)

Figure 6: **Qualitative Comparison.** We show the detection results on a KITTI validation scene by P-RCNN with different input point clouds. We visualize them from both frontal-view images and bird's-eye view (BEV) point maps. Ground-truth boxes are in green and predicted bounding boxes are in red. The observer is at the left-hand side of the BEV map looking to the right. In other words, ground truth boxes on the right are more faraway (i.e., deeper) from the observer, and hence hard to localize. Best viewed in color.

**Qualitative visualization.** In Figure 6, we show an qualitative comparison of detection results on a randomly chosen scene in the KITTI object validation set, using P-RCNN (with confidence > 0.95) with different input signals. Specifically, we show the results from the frontal-view images and the bird's-eye view (BEV) point clouds. In the BEV map, the observer is on the left-hand side looking to

---

than the depth loss (for PIXOR* and AVOD). Nevertheless, Table 3 and Table 6 both suggest the compatibility of the two approaches: combining them leads to the best performance.

the right. It can be seen that the point clouds generated by PSEUDO-LiDAR ++ (SDN alone or SDN +GDC) align better with LiDAR than that generated by PSEUDO-LiDAR (PSMNET). For nearby objects (i.e., bounding boxes close to the left in the BEV map), we see that P-RCNN with any point cloud performs fairly well in localization. However, for faraway objects (i.e., bounding boxes close to the right), PSEUDO-LiDAR with depth estimated from PSMNET predicts objects (red boxes) that are deviated from the ground truths (green boxes). Moreover, the noisy PSMNET points also leads to false negatives. In contrast, the detected boxes by our PSEUDO-LiDAR ++, either with SDN alone or with SDN +GDC, align pretty well with the ground truth boxes, justifying our targeted improvement in estimating faraway depths.

**Additional results, analyses, qualitative visualization and discussions.** We provide results of PSEUDO-LiDAR ++ with fewer LiDAR beams, comparisons to depth completion methods, analysis on depth quality and detection accuracy, run time, failure cases, and more qualitative results in the appendix. With simple optimizations, GDC runs in 90 ms/frame using a single GPU (7.7 ms for KD-tree construction and search).

## 6 CONCLUSION

In this paper we made two contributions to improve the 3D object detection in autonomous vehicles without expensive LiDAR. First, we identify the disparity estimation as a main source of error for stereo-based systems and propose a novel approach to learn depth directly end-to-end instead of through disparity estimates. Second, we advocate that one should not use expensive LiDAR sensors to learn the local structure and depth of objects. Instead one can use commodity stereo cameras for the former and a cheap sparse LiDAR to correct the systematic bias in the resulting depth estimates. We provide a novel graph propagation algorithm that integrates the two data modalities and propagates the sparse yet accurate depth estimates using two sparse matrix solvers. The resulting system, PSEUDO-LiDAR ++ (SDN + GDC), performs almost on par with 64-beam LiDAR systems for $75,000 but only requires 4 beams and two commodity cameras, which could be obtained with a total cost of less than $1,000.

## ACKNOWLEDGMENTS

This research is supported by grants from the National Science Foundation NSF (III-1618134, III-1526012, IIS-1149882, IIS-1724282, and TRIPODS-1740822), the Office of Naval Research DOD (N00014-17-1-2175), the Bill and Melinda Gates Foundation, and the Cornell Center for Materials Research with funding from the NSF MRSEC program (DMR-1719875). We are thankful for generous support by Zillow and SAP America Inc.

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

# Appendix

We provide details omitted in the main text.

- Appendix A: details on constructing the depth cost volume (section 3 of the main paper).
- Appendix B: detailed implementation of the GDC algorithm (section 4 of the main paper).
- Appendix C: additional details of experimental setups (subsection 5.1 of the main paper).
- Appendix D: additional results, analyses, and discussions (subsection 5.2 of the main paper).

## A   DEPTH COST VOLUME

With Equation 2, we know where each grid $(u, v, z)$ in $C_{\text{depth}}$ corresponds to in $C_{\text{disp}}$ (may not be on a grid). We can then obtain features for each grid in $C_{\text{depth}}$ (i.e., $C_{\text{depth}}(u, v, z, :)$) by bilinear interpolation over features on grids of $C_{\text{disp}}$ around the non-grid location (i.e., $\left(u, v, \dfrac{f_U \times b}{z}\right)$). We applied the "grid_sample" function in PyTorch for bilinear interpolation.

We use PSMNET (Chang & Chen, 2018) as the backbone for our stereo depth estimation network (SDN). The only change is to construct the depth cost volume before performing 3D convolutions.

## B   GRAPH-BASED DEPTH CORRECTION (GDC) ALGORITHM

Here we present the GDC algorithm in detail (see algorithm 1). The two steps described in the main paper can be easily turned into two (sparse) linear systems and then solved by using Lagrange multipliers. For the first step (i.e., Equation 7), we solve the same problem as in the main text but we switch the objective to minimizing the $L_2$-norm of $W$ and set $Z - WZ = 0$ as a constraint[5]. For the second step (i.e., Equation 8), we use the *Conjugate Gradient* (CG) to iteratively solve the sparse linear system.

---

**Algorithm 1:** Graph-based depth correction (GDC). ";" stands for column-wise concatenation.

---

**Input:** Stereo depth map $Z \in \mathbb{R}^{(n+m) \times 1}$, the corresponding pseudo-LiDAR (PL) point cloud
$\quad\quad P \in \mathbb{R}^{(n+m) \times 3}$, and LiDAR depths $G \in \mathbb{R}^{n \times 1}$ on the first the $n$ pixels.
**Output:** Corrected depth map $Z' \in \mathbb{R}^{(n+m) \times 1}$
**function** GDC($Z, P, G, K$)
$\quad$ Solve: $W = \arg\min_{W \in \mathbb{R}^{(n+m) \times (n+m)}} \|W\|^2$
$\quad\quad\quad$ s.t. $\quad Z - W \cdot Z = 0,$
$\quad\quad\quad\quad\quad W_{ij} = 0$ if $j \notin \mathcal{N}_i$ (i.e., the set of neighbors of the $i^{th}$ point) according to $P$,
$\quad\quad\quad\quad\quad \sum_j W_{ij} = 1$ for $\forall i = 1, \ldots, n + m.$
$\quad$ Solve: $Z'_{PL} = \arg\min_{Z'_{PL} \in \mathbb{R}^{m \times 1}} \|[G; Z'_{PL}] - W[G; Z'_{PL}]\|^2$
$\quad$ **return** $[G; Z'_{PL}]$
**end**

---

## C   EXPERIMENTAL SETUP

### C.1   SPARSE LiDAR GENERATION

In this section, we explain how we generate sparser LiDAR with fewer beams from a 64-beam LiDAR point cloud from KITTI dataset in detail. For every point $(x_i, y_i, z_i) \in \mathbb{R}^3$ of the point cloud in one

---

[5]These two problems yield identical solutions but we found the second one is easier to solve in practice. We note that, Equation 7 is an under-constrained problem, with infinitely many solutions. To identify a single solution, we add a small $L_2$ regularization term to the objective (as mentioned in the main text).

scene (in LiDAR coordinate system ($x$: front, $y$: left, $z$: up, and $(0, 0, 0)$ is the location of the LiDAR sensor)), we compute the elevation angle $\theta_i$ to the LiDAR sensor as

$$\theta_i = \arg\cos\left(\frac{\sqrt{x_i^2 + y_i^2}}{\sqrt{x_i^2 + y_i^2 + z_i^2}}\right).$$

We order the points by their elevation angles and slice them into separate lines by step $0.4°$, starting from $-23.6°$ (close to the Velodyne 64-beam LiDAR SPEC). We select LiDAR points whose elevation angles fall within $[-2.4°, -2.0°) \cup [-0.8°, -0.4°)$ to be the 2-beam LiDAR signal, and similarly $[-2.4°, -2.0°) \cup [-1.6°, -1.2°) \cup [-0.8°, -0.4°) \cup [0.0°, 0.4°)$ to be the 4-beam LiDAR signal. We choose them in such a way that consecutive lines has a $0.8°$ interval, following the SPEC of the "cheap" 4-beam LiDAR ScaLa. We visualize these sparsified LiDAR point clouds from the bird's-eye view on one example scene in Figure 7.

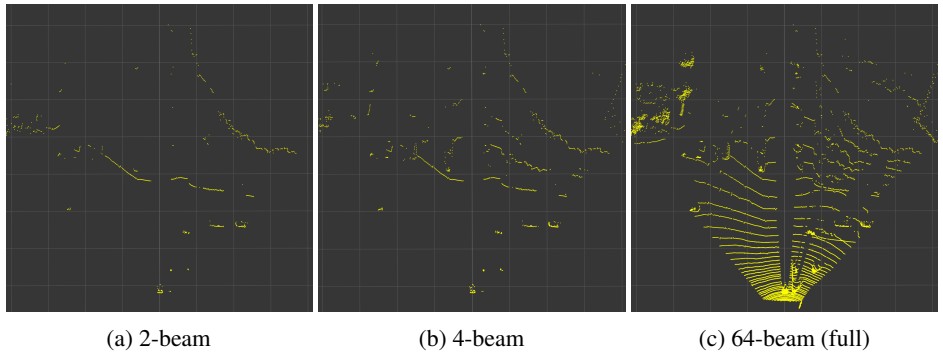

(a) 2-beam        (b) 4-beam        (c) 64-beam (full)

Figure 7: **Bird's-eye views of sparsified LiDAR on an example scene.** The observer is on the bottom side looking up. We filter out points invisible from the left image. (One floor square is 10m × 10m.)

### C.2 3D OBJECT DETECTION ALGORITHMS

In this section, we provide more details about the way we train 3D object detection models on pseudo-LiDAR point clouds. For AVOD, we use the same model as in (Wang et al., 2019a). For P-RCNN, we use the implementation provided by the authors. Since the P-RCNN model exploits the sparse nature of LiDAR point clouds, when training it with pseudo-LiDAR input, we will first sparsify the point clouds into 64 beams using the method described in subsection C.1. For PIXOR⋆, we implement the same base model structure and data augmentation specified by Yang et al. (2018b), but without the "decode fine-tune" step and focal loss. Inspired by the trick in (Liang et al., 2018), we add another image feature (ResNet-18 by He et al. (2016)) branch along the LiDAR branch, and concatenate the corresponding image features onto the LiDAR branch at each stage. We train PIXOR⋆ using RMSProp with momentum $0.9$, learning rate $10^{-5}$ (decay by 10 after 50 and 80 epochs) for 90 epochs. The BEV evaluation results are similar to the reported results (see Table 1).

## D ADDITIONAL RESULTS, ANALYSES, AND DISCUSSIONS

### D.1 ABLATION STUDY

In Table 6 and Table 7 we provide more experimental results aligned with experiments in subsection 5.2 of the main paper. We conduct the same experiments on two other models, AVOD and PIXOR⋆, and observe similar trends of improvements brought by learning with the depth loss (from PSMNET to PSMNET +DL), constructing the depth cost volume (from PSMNET +DL to SDN), and applying GDC to correct the bias in stereo depth estimation (comparing SDN +GDC with SDN).

We note that, in Table 7, results of AVOD (or PIXOR⋆) with SDN + L# are worse than those with L# at the moderate and hard settings. This observation is different from that in Table 4, where P-RCNN with SDN + L# outperforms P-RCNN with L# in 5 out of 6 comparisons. We hypothesize that this is because P-RCNN takes sparsified inputs (see subsection C.2) while AVOD and PIXOR⋆ take dense inputs. In the later case, the four replaced LiDAR beams in SDN + L# will be dominated by the dense stereo depths so that SDN + L# is worse than L#.

## D.2 USING FEWER LiDAR BEAMS

In PL++ (i.e., SDN + GDC), we use 4-beam LiDAR to correct the predicted point cloud. In Table 8, we investigate using fewer (and also potentially cheaper) LiDAR beams for depth correction. We observe that even with 2 beams, GDC can already manage to combine the two signals and yield a better performance than using 2-beam LiDAR or pseudo-LiDAR alone.

Table 6: **Ablation study on stereo depth estimation.** We report $AP_{BEV}$ / $AP_{3D}$ (in %) of the **car** category at IoU= 0.7 on the KITTI validation set. DL stands for depth loss.

| Depth Estimation | PIXOR* | | | AVOD | | |
|---|---|---|---|---|---|---|
| | Easy | Moderate | Hard | Easy | Moderate | Hard |
| PSMNET | 73.9 / - | 54.0 / - | 46.9 / - | 74.9 / 61.9 | 56.8 / 45.3 | 49.0 / 39.0 |
| PSMNET + DL | 75.8 / - | 56.2 / - | 51.9 / - | 75.7 / 60.5 | 57.1 / 44.8 | 49.2 / 38.4 |
| SDN | 79.7 / - | 61.1 / - | 54.5 / - | 77.0 / 63.2 | 63.7 / 46.8 | 56.0 / 39.8 |

Table 7: **Ablation study on leveraging sparse LiDAR.** We report $AP_{BEV}$ / $AP_{3D}$ (in %) of the **car** category at IoU= 0.7 on the KITTI validation set. L# stands for 4-beam LiDAR signal. SDN +L# means we replace the depth of a portion of pseudo-LiDAR points (i.e., landmark pixels) by L#.

| Depth Estimation | PIXOR* | | | AVOD | | |
|---|---|---|---|---|---|---|
| | Easy | Moderate | Hard | Easy | Moderate | Hard |
| SDN | 79.7 / - | 61.1 / - | 54.5 / - | 77.0 / 63.2 | 63.7 / 46.8 | 56.0 / 39.8 |
| L# | 72.0 / - | 64.7 / - | 63.6 / - | 77.0 / 62.1 | 68.8 / 54.7 | 67.1 / 53.0 |
| SDN + L# | 75.6 / - | 59.4 / - | 53.2 / - | 84.1 / 66.0 | 67.0 / 53.1 | 58.8 / 46.4 |
| SDN + GDC | 84.0 / - | 71.0 / - | 65.2 / - | 86.8 / 70.7 | 76.6 / 56.2 | 68.7 / 53.4 |

Table 8: **Ablation study on the sparsity of LiDAR.** We report $AP_{BEV}$ / $AP_{3D}$ (in %) of the **car** category at IoU= 0.7 on the KITTI validation set. L# stands for using sparse LiDAR signal alone. The number in brackets indicates the number of beams in use.

| Depth Estimation | P-RCNN | | | PIXOR* | | |
|---|---|---|---|---|---|---|
| | Easy | Moderate | Hard | Easy | Moderate | Hard |
| SDN | 82.0 / 67.9 | 64.0 / 50.1 | 57.3 / 45.3 | 79.7 / - | 61.1 / - | 54.5 / - |
| L# (2) | 69.2 / 46.3 | 62.8 / 41.9 | 61.3 / 40.0 | 66.8 / - | 55.5 / - | 53.3 / - |
| L# (4) | 73.2 / 56.1 | 71.3 / 53.1 | 70.5 / 51.5 | 72.0 / - | 64.7 / - | 63.6 / - |
| SDN + GDC (2) | 87.2 / 73.3 | 72.0 / 56.6 | 67.1 / 54.1 | 82.0 / - | 65.3 / - | 61.7 / - |
| SDN + GDC (4) | 88.2 / 75.1 | 76.9 / 63.8 | 73.4 / 57.4 | 84.0 / - | 71.0 / - | 65.2 / - |

Table 9: **Comparison of GDC and PNP for 3D object detection.** We report $AP_{BEV}$ / $AP_{3D}$ (in %) of the **car** category at IoU= 0.7 on the KITTI validation set, using SDN + PNP or SDN + GDC for depth estimation and P-RCNN or PIXOR* for detection.

| Input signal | P-RCNN | | | PIXOR* | | |
|---|---|---|---|---|---|---|
| | Easy | Moderate | Hard | Easy | Moderate | Hard |
| SDN + PNP | 86.3 / 72.1 | 73.3 / 58.9 | 67.2 / 54.2 | 79.1 / - | 64.2 / - | 54.0 / - |
| SDN + GDC | 88.2 / 75.1 | 76.9 / 63.8 | 73.4 / 57.4 | 84.0 / - | 71.0 / - | 65.2 / - |

## D.3 DEPTH CORRECTION VS. DEPTH COMPLETION

We compare our GDC algorithm for depth correction to depth completion algorithms, which aim to "densify" LiDAR data beyond the beam lines (Wang et al., 2018; Tomasello et al., 2018; Ma et al., 2019; Yang et al., 2019; Cheng et al., 2018; Torres-Mendez & Dudek, 2004)[6]. We note that most depth completion approaches take as input a 64-beam LiDAR and a single image, while our focus is on fusing a much sparser 4-beam LiDAR and stereo depths. As such, the two problems are not

---

[6]Torres-Mendez & Dudek (2004) use MRFs and may thus require less (or even no) training data compared to deep learning algorithms: a property shared by GDC.

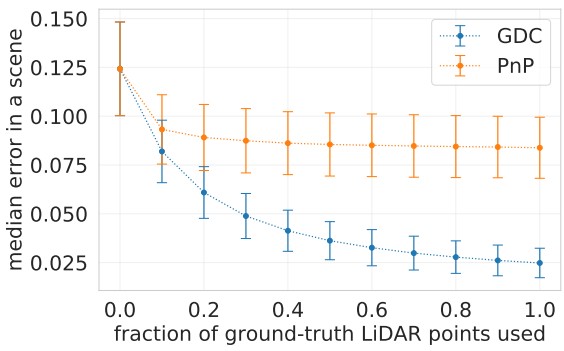
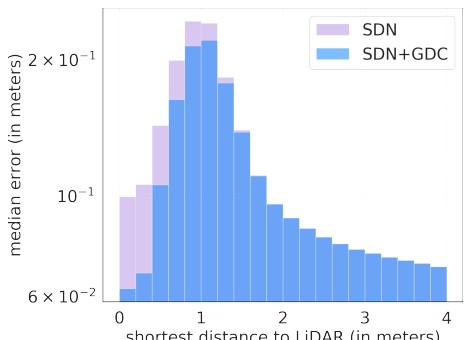

Figure 8: **Comparison of GDC and PNP for depth correction.** We report the median of absolute errors on the KITTI validation set. See text for details.

Figure 9: **Median depth estimation errors w.r.t. the shortest distances to 4-beam LiDAR points on KITTI validation set.**

Table 10: **Comparison of 3D object detection using the naive and optimized implementation of GDC.** We report $AP_{BEV}$ / $AP_{3D}$ (in %) of the **car** category at IoU$= 0.7$ on the KITTI validation set, using P-RCNN for detection.

|          | Easy        | Moderate    | Hard        |
|----------|-------------|-------------|-------------|
| Naive    | 88.2 / 75.1 | 76.9 / 63.8 | 73.4 / 57.4 |
| Optimized| 87.6 / 75.0 | 76.3 / 63.4 | 73.1 / 57.0 |

commensurate. Also, our GDC algorithm is a general, simple, inference-time approach that *requires no training*, unlike prior learning-based approaches to depth completion.

Here we empirically compare to PNP (Wang et al., 2018), a recently proposed depth completion algorithm compatible with any (even stereo) depth estimation network, similar to GDC. We use SDN for initial depth estimation, and evaluate GDC and PNP by randomly selecting a fraction of LiDAR points as provided ground truths and calculating the median absolute depth errors on the remaining LiDAR points. As shown in Figure 8, GDC outperforms PNP by a large margin. Table 9 shows a further comparison to PNP on 3D object detection. We apply PNP and GDC respectively to correct the depth estimates obtained from SDN, train a P-RCNN or PIXOR$^\star$ using the resulting pseudo-LiDAR points on the KITTI training set, and compare the detection results on the KITTI validation set. In either case, SDN + GDC outperforms SDN + PNP by a notable margin.

## D.4   RUN TIME

With the following optimizations for implementation,

1. Sub-sampling pseudo-LiDAR points: keeping at most one point within a cubic of size $0.1m^3$
2. Limiting the pseudo-LiDAR points for depth correction: keeping only those whose elevation angles are within $[-3.0°, 0.4°)$ (the range of 4-beam LiDAR plus $0.6°$; see subsection C.1 for details)
3. After performing GDC for depth correction, combining the corrected pseudo-LiDAR points with those outsides the elevation angles of $[-3.0°, 0.4°)$

GDC runs in 90 ms/frame using a single GPU (7.7ms for KD-tree construction and search, 46.5ms for solving $W$, and 26.9ms for solving $Z'_{PL}$) with negligible performance difference (see Table 10). For consistency, all results reported in the main paper are based on the naive implementation. Further speedups can be achieved by CUDA programming for GPUs.

## D.5   STEREO DEPTH VS. DETECTION

We quantitatively evaluate the stereo depths by median errors in Figure 4 of the main text (numerical values are listed in Table 11). In Table 12 we further show mean errors with standard deviation (the large standard deviation likely results from outliers such as occluded pixels around object boundaries).

Table 11: **Median depth estimation errors over various depth ranges (numerical values of Figure 4).**

| Signal | range (m) | | | | | | |
|---|---|---|---|---|---|---|---|
| | 0-10 | 10-20 | 20-30 | 30-40 | 40-50 | 50-60 | 60-70 |
| PSMNet | 0.04 | 0.11 | 0.36 | 0.83 | 1.24 | 1.98 | 2.43 |
| SDN | 0.07 | 0.12 | 0.30 | 0.60 | 0.89 | 1.31 | 1.73 |
| SDN + GDC | 0.07 | 0.12 | 0.27 | 0.51 | 0.74 | 1.03 | 1.53 |

Table 12: **Mean depth estimation errors (with standard deviation) over various depth ranges.**

| Signal | range (m) | | | | | | |
|---|---|---|---|---|---|---|---|
| | 0-10 | 10-20 | 20-30 | 30-40 | 40-50 | 50-60 | 60-70 |
| PSMNet | 0.18±0.93 | 0.36±1.20 | 0.97±2.32 | 2.02±4.05 | 2.94±5.64 | 4.61±8.03 | 6.03±10.32 |
| SDN | 0.21±0.89 | 0.35±1.16 | 0.87±2.31 | 1.80±4.22 | 2.67±6.00 | 4.27±8.78 | 5.82±11.23 |
| SDN + GDC | 0.21±0.90 | 0.35±1.17 | 0.84±2.34 | 1.74±4.27 | 2.59±6.06 | 4.14±8.85 | 5.72±11.29 |

For both tables, we divide pixels into beams according to their truth depths, and evaluate on pixels not on the 4-beam LiDAR. The improvement of SDN (+ GDC) over PSMNET becomes larger as we consider pixels farther away. Table 13 further demonstrates the relationship between depth quality and detection accuracy: SDN (+ GDC) significantly outperforms PSMNET for detecting faraway cars. We note that, for very faraway cars (i.e., 50-70 m), the number of training object instances are extremely small, which suggests that the very poor performance might partially cause by over-fitting.

Further, we apply the same evaluation procedure but group the errors by the shortest distance between each PSEUDO-LIDAR point and the 4-beam LiDAR points in Figure 9. We can see that the closer the PSEUDO-LIDAR points are to the 4-beam LiDAR points, the bigger improvement GDC can bring.

### D.6 CONNECTED COMPONENTS IN KNN GRAPHS OF PSEUDO-LIDAR POINTS BY SDN

Here, we provide empirical analysis on the relationship between the $k$ we choose in building the K-nearest-neighbor graph of PSEUDO-LIDAR points by SDN and the number of connected components of that graph. We show the results on KITTI validation set in Figure 11. It can be seen that with $k \geq 9$, the average number of connected components in the graph is smaller than 2.

### D.7 FAILURE CASES AND WEAKNESS

There is still a gap between our approach and LiDAR for faraway objects (see Table 13). We further analyze $AP_{BEV}$ at different IoU in Figure 10. For low IoU (0.2-0.5), SDN (+GDC) is on par with LiDAR, but the gap increases significantly at high IoU thresholds. This suggests that the predominant gap between our approach and LiDAR is because of mislocalization, perhaps due to residual inaccuracies in depth.

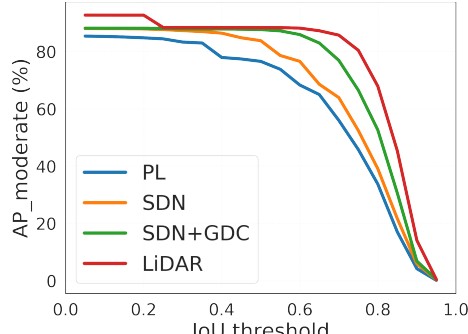

Figure 10: **IoU vs. $AP_{BEV}$ on KITTI validation set on the car category (moderate).**

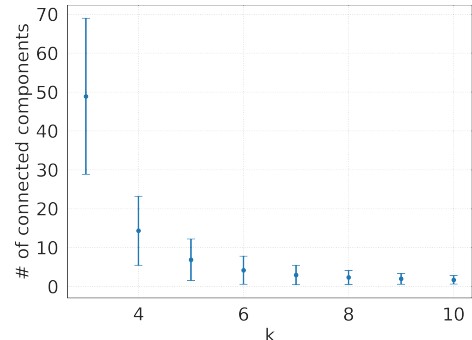

Figure 11: $k$ **vs. average number of connected components in KNN graphs of PSEUDO-LIDAR points by SDN.**

Table 13: **3D object detection at various depth ranges.** We compare different input signals. We report AP$_{BEV}$ / AP$_{3D}$ (in %) of the **car** category at IoU= 0.7 on the KITTI validation set, using P-RCNN for detection. In the last two rows we show the number of car objects in KITTI object train and validation sets within different ranges.

| Input signal | 0-30 m | 30-50 m | 50-70 m |
|---|---|---|---|
| PSMNET | 65.6 / 54.0 | 15.8 / 6.9 | 0.0 / 0.0 |
| SDN | 68.6 / 56.7 | 27.4 / 11.3 | 0.7 / 0.0 |
| SDN + GDC | 84.7 / 67.8 | 49.9 / 31.5 | 2.5 / 1.0 |
| LiDAR | 88.5 / 84.0 | 69.9 / 51.5 | 8.9 / 3.4 |
| # OBJECTS-TRAIN | 6903 | 3768 | 76 |
| # OBJECTS-VAL | 7379 | 3542 | 39 |

## D.8 QUALITATIVE RESULTS

In Figure 6,12,13 and Figure 14, we show detection results using P-RCNN (with confidence > 0.95) with different input signals on four randomly chosen scenes in the KITTI object validation set. Specifically, we show the results from the frontal-view images and the bird's-eye view (BEV) point clouds. In the BEV map, the observer is on the left-hand side looking to the right. It can be seen that the point clouds generated by PSEUDO-LIDAR ++ (SDN alone or SDN +GDC) align better with LiDAR than those generated by PSEUDO-LIDAR (PSMNET). For nearby objects (i.e., bounding boxes close to the left in the BEV map), we see that P-RCNN with any point cloud performs fairly well in localization. However, for faraway objects (i.e., bounding boxes close to the right), PSEUDO-LIDAR with depth estimated from PSMNET predicts objects (red boxes) deviated from the ground truths (green boxes). Moreover, the noisy PSMNET points also leads to several false positives or negatives. In contrast, the detected boxes by our PSEUDO-LIDAR ++, either with SDN alone or with SDN +GDC, align pretty well with the ground truth boxes, justifying our targeted improvement in estimating faraway depths. In Figure 12, we see one failure case for both PSEUDO-LIDAR and PSEUDO-LIDAR ++: the most faraway car is missed, while LiDAR signal can still detect it, suggesting that for very faraway objects stereo-based methods may still have limitation.

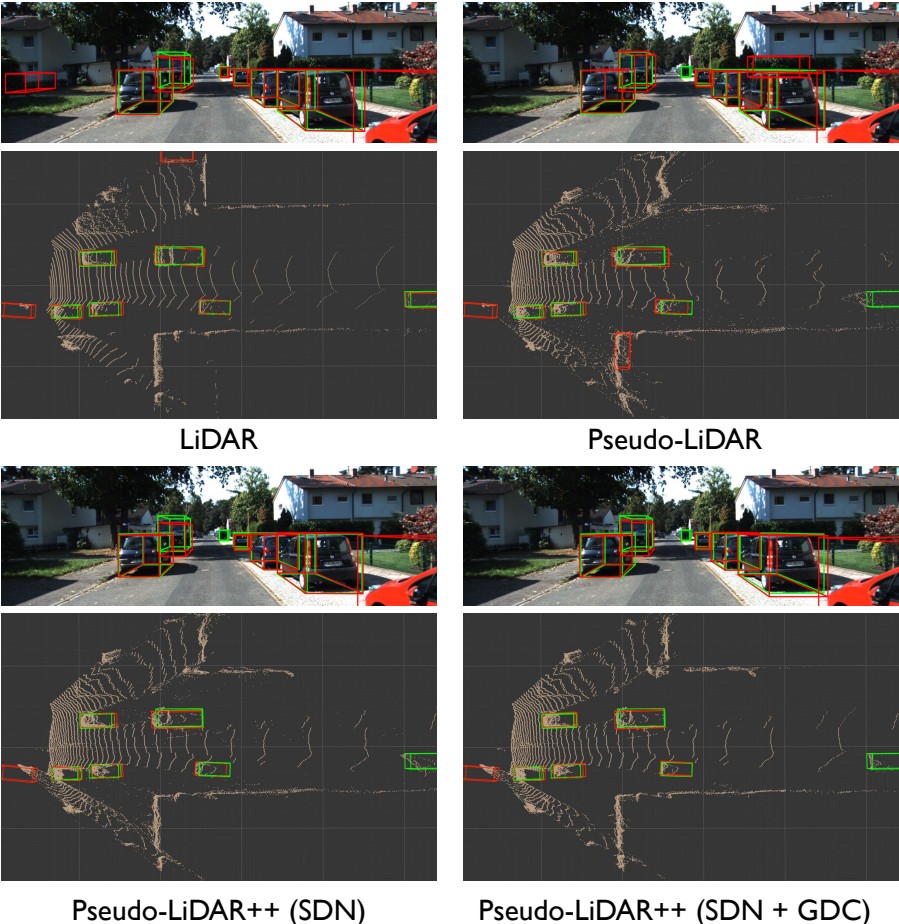

Figure 12: **Qualitative Comparison.** We show the detection results on a KITTI validation scene by P-RCNN with different input point clouds. We visualize them from both frontal-view images and bird's-eye view (BEV) point maps. Ground-truth boxes are in green and predicted bounding boxes are in red. The observer is at the left-hand side of the BEV map looking to the right. In other words, ground truth boxes on the right are more faraway (i.e., deeper) from the observer, and hence hard to localize. Best viewed in color.

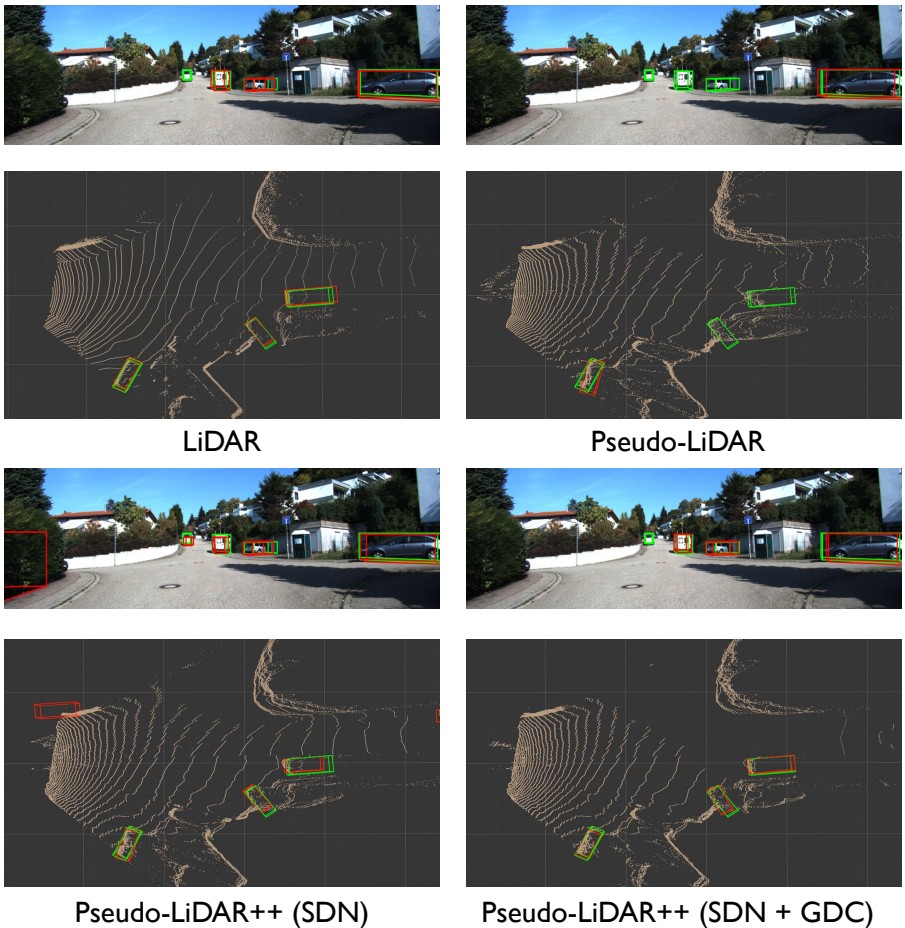

LiDAR · Pseudo-LiDAR

Pseudo-LiDAR++ (SDN) · Pseudo-LiDAR++ (SDN + GDC)

Figure 13: **Qualitative Comparison - another example.** The same setup as in Figure 12

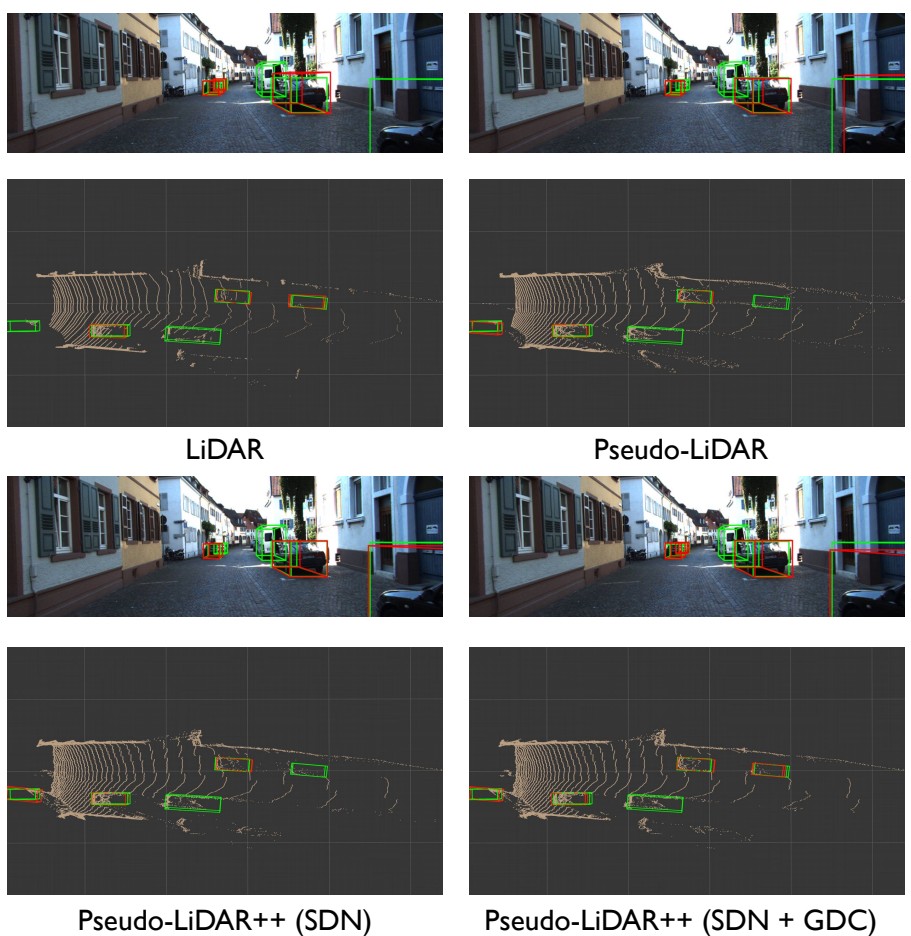

Figure 14: **Qualitative Comparison - another example.** The same setup as in Figure 12

