# OpenReview forum: "Pseudo-LiDAR++: Accurate Depth for 3D Object Detection in Autonomous Driving"
_ICLR.cc/2020/Conference — Accept (Poster)_

### Official Review · AnonReviewer3 · 2019-10-22
**Official Blind Review #3**

**Rating:** 6

**Review:**

The paper proposes two extensions to the recent work of (Wang et al., 2019) on 3D object detection with pseudo LiDAR data. Wang et al. showed that 3D object detection using stereo images as inputs can be significantly improved if the depth map is projected to 3D and treated like a LiDAR point cloud (i.e., using methods that utilize the LiDAR point cloud). This paper shows that one shortcoming of this approach is given by the fact that the depth uncertainty increases the farther the objects are away. To remedy this, the authors propose to train the stereo estimation network (based on Chang & Chen, 2018) directly with depth outputs, instead of disparity values (inverse depth), by rewriting the loss and converting the cost volume. This already boosts the performance for far away objects. The authors demonstrate that the usage of a (simulated) low-cost 4-beam LiDAR can further facilitate the detection. For this purpose a graph diffusion algorithm is listed that aligns the pseudo LiDAR point cloud from the stereo set-up with the depth estimates from the low-cost LiDAR. Simulating the low-cost LiDAR on the Kitti benchmarks shows that this approach further increases the performance of the object detection methods.

In general, I am in favour of accepting the paper as it shows two orthogonal and interesting additions to the pseudo LiDAR paper of Wang et al. that improve its performance. However, I would like to see some clarifications in the rebuttal.

The proposed stereo network converts a disparity cost volume to a depth cost volume using bilinear interpolation. I agree, that the 3D convolutions are more meaningful (given the spacing of the grid cells) on the latter, but why the detour over the disparity cost volume? It should be possible to build the depth cost volume directly, which would lead to decreased memory consumption and speed up the method without any loss in accuracy?

One assumption of the second contribution (GDC) is that at least one beam of the LiDAR will hit the k-connected local point cloud. Can you give some bounds on the likelihood that this happens, especially for far away objects it could be unlikely, although it is most beneficial for those objects.
Further, I am missing a details on the optimization of (7) and (8). What is meant with slight L2 regularization? In the appendix it is also stated that a slightly different objective is optimized?
Finally, the notation could also be improved. The authors are using L and G for the LiDAR point cloud and PL and Z for the pseudo LiDAR point cloud and then in the Z' is used for both.

Fig. 4 shows the median error in meters for the different variants of the stereo network. Why has the median been used? Are there severe outliers? If yes, it would also be interesting to quantify those and compare them (e.g., box plots).

In the abstract and in the discussion the authors oversell their results a bit. At the one hand they state that PL++ with GDC performs significantly better than PL++ w/o GDC, on the other hand they also claim that PL++ achieves comparable results to models that have access to the full 64-beam LiDAR data. However, if you compare the differences, then the gaps are for several cases almost as big, or bigger as in the former claim.

Things to improve the paper that did not impact the score:
- In equation (2) you could replace the x with a . (\cdot), or completely remove it
- On page 5: KNN neighbors -> k-nearest neighbors
- Also on page 5: write out W.l.o.g.
- In Tab. 1 it would help to highlight (bold) the best entries per column

**Experience Assessment:**

I have published one or two papers in this area.

**Review Assessment: Checking Correctness Of Derivations And Theory:**

N/A

**Review Assessment: Checking Correctness Of Experiments:**

I carefully checked the experiments.

**Review Assessment: Thoroughness In Paper Reading:**

I read the paper thoroughly.

---

> ### Author Response · Authors · 2019-11-14
> **We thank the reviewer for the positive review and constructive comments!**
>
> 1.[detour over the disparity cost volume] Thanks for pointing this out. It is definitely possible to construct the depth cost volume directly, however constructing the disparity cost volume brings us simplicity in implementation and efficiency through utilizing matrix operations. Although it does require some additional GPU memory, it is on the order of a few hundred megabytes. In terms of computation, the most costly part of the depth estimation model is 3D convolutions on the depth cost volume. In comparison, the memory and computation cost of constructing the disparity cost volume first is quite small.
>
> 2.[analysis on GDC] Thanks for mentioning your concern about GDC with regards to lasers missing connected components. In practice, we have not observed that this is a problem, in part because the pseudo-Lidar point cloud is sufficiently dense, and we choose k to be large enough (k=10) that the graph is typically connected (or consists of few large connected components). We will add a more detailed discussion in the final paper about this issue and provide empirical numbers for various values of k.
>
> 3. [notation and optimization on equation (7) and (8)] Thanks for pointing out the notational collisions, we will correct these in the final version.
>
> 4. [L2-regularization] The first step of GDC, i.e., equation (7), is an under-constrained problem, with infinitely many solutions. To identify a single solution, we add a small L2 regularization term to the objective (main paper). In the appendix, we switch the objective with the constraints by minimizing the L2-norm of W and set Z-WZ=0 as a constraint. These two problems yield identical solutions but we found the first formulation easier to explain (i.e., adding a regularizer to equation (7)) while the second one is easier to solve in practice. We apologize for the confusion and will clarify our description in the final version.
>
> 5.[median error] Thanks for the suggestion. We re-evaluate depth estimation with mean error and find that it is larger than the median error, which likely results from outliers such as occluded pixels around object boundaries. We list the mean error and the standard deviation below. SDN+GDC still achieves the lowest mean error (except for 0-10 meters), followed by SDN and then the vanilla disparity-based PSMNet. We will include a more detailed box-plot graph in the final version.
> mean\range  0-10   10-20   20-30   30-40   40-50   50-60   60-70
> PL         0.176   0.359   0.967   2.023   2.936   4.611   6.025
> SDN        0.212   0.352   0.865   1.799   2.668   4.272   5.824
> SDN+GDC    0.209   0.345   0.842   1.744   2.590   4.137   5.721
>
> std\range   0-10   10-20   20-30   30-40   40-50   50-60   60-70
> PL         0.929   1.200   2.320   4.049   5.641   8.034   10.317
> SDN        0.894   1.157   2.310   4.218   6.004   8.776   11.232
> SDN+GDC    0.897   1.167   2.338   4.266   6.058   8.852   11.293
>
> 6.[Overselling results] We apologize; we did not mean to oversell our results. This is an unfortunate naming conflict. In the abstract, by PL++ we mean SDN+GDC, while in Table 1, we separate SDN and GDC for analysis but still call them both PL++. With SDN+GDC, our model can achieve comparable results to models that have access to the full 64-beam LiDAR data on some of the metrics (as mentioned in section 5.2 and the introduction). We will clarify this in the final version.

---

### Official Review · AnonReviewer1 · 2019-10-24
**Official Blind Review #1**

**Rating:** 6

**Review:**

The paper proposes to improve the idea of using stereo + lidar -style object detection to form stereo-based 3D object detection, building off of pseudo-lidar. In particular, it proposes to (a) switch the loss function for stereo deep networks from disparity to depth (b) do the stereo cost volume analysis in a depth volume (via resampling) rather than disparity volume and (c) if sparse lidar is available, align the estimated depth with the sparse lidar. Each seems to improve results, and the resulting system achieves good results on KITTI and outperforms past work in this area.

Positives:
+The paper proposes three ideas that seem good and lead to improvements that are demonstrated empirically.
+The paper is well written
+The experiments are exceptionally thorough
+The ideas seems to me to be of obvious importance, although I realize that I'm likely not qualified to make a statement about this, and this should perhaps be done by a roboticist.

Negatives:
-Most of the heavy lifting in the case without sparse LIDAR is done by tweaking the loss function rather than the cost, although the remaining gain is still pretty good
-I am not sure if this is a negative, but this is really a 3D vision paper. I do some form of 3D vision, but I really don't feel confident about my ability to assess whether doing stereo matching in a depth cost volume as opposed to disparity is correct -- I really haven't worked on stereo. It seems to work well, but I feel as if the wrong people are being asked to review the paper. I leave questions of venue to the area chair though.

Overall, I am inclined to accept the paper. I am a tiny bit worried about venue and whether the right people will check the work, but I don't think this should be decided by reviewers. However, I think the experiments are quite thorough and the paper is clearly above the bar.

In more detail:

Method:
+The method reads quite well and the idea is clean. I particularly like the graph-based depth correction algorithm, and the LLE-like way of adjusting the estimated depthmap. I have a few small comments below that do not affect my judgment, but I think would improve the paper.
= Small thought: the words systematic bias throughout is primarily referring to a bias for a particular object as opposed to a bias of the system (i.e., any individual object is too far or too close). This seems non-standard to me. A systematic bias would be that everything's too far away by 1m for instance.

Experiments:
+The experiments are exceptionally thorough, and of my pile of ICLR papers, this by far has the most thorough and well-thought-out experimental analysis.
+The system shows systematic improvements on 3 different LIDAR-based object detection systems; I think this is great.
=I'm not sure whether the 64-beam LIDAR can be subsampled to imitate 4-beam LIDAR. I simply don't know enough about the hardware to know if this is a sensible approximation.
-Table 3 primarily suggests that the vast majority of the hard work in the non-sparse LIDAR is done by the depth loss rather than the depth cost volume. The resulting change is still pretty good (although I suspect that if you stuck in a coordconv in the disparity cost volume, it would handle the fact that you want unequal smoothing).
-The burial of the results on depth prediction results in the appendix with one is a little surprising as is the solitary table on it, but I understand the need to focus on 3D detection.

Small stuff that doesn't affect my review:
1) Framing the problem as having ethical considerations is, in my view, not necessary -- should all network compression papers start arguing that it is of profound ethical importance to figure out your bit quantization?
2) Last paragraph above Section 4 "gird" -> "Grid"
3) Figure 3 caption "pruple" -> "purple"
4) Figure 4 is suboptimal -- I assume SDN+GDC < SDN < Disparity Net, but this is hard to verify.
5) Calling the network "Disparity Net" is a bit of an issue given that there's DispNet already
6) "Figure 1 illustrates beautifully how" -> Please don't editorialize like this

-----------------------

Post rebuttal update: I have read the rebuttal and maintain my belief that the paper should be accepted.

**Experience Assessment:**

I have read many papers in this area.

**Review Assessment: Checking Correctness Of Derivations And Theory:**

I assessed the sensibility of the derivations and theory.

**Review Assessment: Checking Correctness Of Experiments:**

I assessed the sensibility of the experiments.

**Review Assessment: Thoroughness In Paper Reading:**

I read the paper thoroughly.

---

> ### Author Response · Authors · 2019-11-14
> **We thank the reviewer for the positive review and constructive comments!**
>
> 1. [Depth cost volume] We would like to point out that the gain depends on the 3D detector in use. Table 3 suggests that the depth loss provides more gain than the depth cost volume (for PointRCNN). In Table 6, we, however, see that the depth cost volume provides comparable or even bigger gain than the depth loss (for PIXOR and AVOD). Nevertheless, Table 3 and Table 6 both suggest the compatibility of the two approaches: combining them leads to the best performance. Applying coordconv is an interesting idea, and if it works, it also supports our claim that smoothing the disparity cost volume with conventional convolutions is inappropriate for depth estimation.
>
> 2. [Other comments] We thank the reviewer for pointing out the typos, which we will correct. We will also search for a better alternative for "Disparity Net" to contrast “Stereo Depth Net”, or clearly indicate that it is not DispNet. We will try to use different colors to clarify Figure 4. We note that Table 11 provides the error values for Figure 4 and we will clearly mention it in the main paper. Section D.3 and Figure 7 also discuss and compare our GDC algorithm to depth completion in terms of the depth estimation error.

---

### Official Review · AnonReviewer2 · 2019-10-24
**Official Blind Review #2**

**Rating:** 6

**Review:**

Summary:
This paper describes a new method for Pseudo-Lidar, that is the reliable recovery of a 3D point cloud from 2D inputs and subsequent detection of 3D objects from the point cloud. The authors focus on improving the accuracy of the reconstructed point cloud by formulating a loss in depth, rather than disparity space, and by using sparse true lidar readings to align estimates. These techniques lead to a boost in 3D object detection performance.

Strengths:
The Pseudo-Lidar method has been well-received, and it appears that this paper makes a nice improvement on the previous in terms of 3D point cloud accuracy. While the image results here are convincing, I would have liked to see an added empirical evaluation of precisely how accurate the resulting 3D reconstructions are, measured against ground truth 3D lidar on a test set. Do the point clouds only look accurate locally (and perhaps near known objects give good shape due to regularity), or are the metric results also quite strong indeed?

I found the author's technical analysis and method description to be clear and well-motivated. None of the math or formulations are entirely surprising, but they are new to this area, so this appeared as nice sensible progress to me.

This area is closely tied to the self-driving car application, and thus bottom-line performance is the key measuring stick for impact on practitioners. For 3D object detection, the main goal of interest, the authors show up to 20% improvements for their combined method over quite recent and strong PL methods (although the new method uses sparse lidar, which is a great advantage, hence not entirely equal comparison). This is the main impact of the paper, as I see it, and enough reason for acceptance.

Areas for Improvement:
I found that the authors did not sufficiently recognize that there have been a wide variety of methods utilizing sparse 3D along with dense 2D images to interpolate to full 3D. For example, [A] is one I recall well from 15 years back, but at that time there was a strong community in this area, so I encourage the authors to do a bit more thorough review.

This paper has the fewest qualitative examples of 3D objects detected among the recent papers I've read. The final pages of the Appendix contain a few more of these visuals, but there are too few in the main paper for the reader to get any intuitive feeling of the physical meaning of your performance improvement. I'd like to see you add several examples, even if small, into the paper to aid in this understanding.

Decision: weak accept due to the nice clear method that gives a strong improvement on an important area to industry today.

[A] Statistical Inference and Synthesis in the Image Domain for Mobile Robot Environment Modeling, Luz Abril Torres-Méndez and Gregory Dudek. In Proceedings of the IEEEE/RSJ/GI International Conference on Intelligent Robots and Systems (IROS), Sendai, Japan, 2004.

**Experience Assessment:**

I have read many papers in this area.

**Review Assessment: Checking Correctness Of Derivations And Theory:**

I assessed the sensibility of the derivations and theory.

**Review Assessment: Checking Correctness Of Experiments:**

I assessed the sensibility of the experiments.

**Review Assessment: Thoroughness In Paper Reading:**

I read the paper at least twice and used my best judgement in assessing the paper.

---

> ### Author Response · Authors · 2019-11-14
> **We thank the reviewer for the positive review and constructive comments!**
>
> 1. [empirical evaluation of 3D reconstructions] Thanks for your question. The point clouds are surprisingly accurate even for points quite far from the 4 laser beams. You can see this to some degree in Figure 4 of the main paper and Table 11 of the appendix. Here, we consider LiDAR points originating from beams beside the selected 4 beams as ground-truth and report the difference between predicted depth and the LiDAR estimates. Some of these beams are quite far from the selected 4. SDN+GDC achieves the lowest depth estimation error, followed by SDN and then the vanilla disparity-based PSMNet. We will highlight this in the final version and add a special table that summarizes the error as a function of distance from the closest LiDAR point.
>
> 2. [literature of depth completion] We thank the reviewer for the comment and the reference. We will include further discussions with regard to [Torres-Méndez and Dudek, 2004], which uses MRFs and may thus require less (or even no) training data compared to deep learning algorithms: a property shared by GDC. In Figure 7 and Table 9, we compare our GDC method with PnP, which is a depth completion method proposed recently and can be applied to stereo images. We will try to add additional comparisons and delve deeper into more classical techniques and include more details in the final version.
>
> 3. [More qualitative results] Thanks for the suggestion, we have updated two more qualitative results in the appendix of the current revised version. We will add more qualitative results in the appendix or even in the main paper if we can create space.

---

### Public Comment · ~Siddharth_Srivastava3 · 2019-10-13
**Relevant paper**

Following paper could be of relevance to your work as well "Learning 2D to 3D Lifting for Object Detection in 3D for Autonomous Vehicles" (to be presented at IROS 2019), https://arxiv.org/pdf/1904.08494.pdf

---

> ### Author Response · Authors · 2019-10-22
> **Thanks**
>
> Thanks for your comment. This looks very interesting! Note that our work is concurrent with this paper, which is yet to come out at IROS. Nevertheless, we will discuss similarities and differences in the camera-ready.
>
> Thanks again for bringing this to our attention! Glad to see that the problem of image-based 3D detection is getting broader attention.

---

### Decision · Program_Chairs · 2019-12-19

**Decision:**

Accept (Poster)

**Comment:**

Three knowledgable reviewers give a positive evaluation of the paper. The decision is to accept.